# First Report of the Ash Sawfly, *Tomostethus nigritus*, Established on *Fraxinus excelsior* in the Republic of Ireland

**DOI:** 10.3390/insects13010006

**Published:** 2021-12-21

**Authors:** Erika Soldi, Emma Fuller, Anna M. M. Tiley, Archie K. Murchie, Trevor R. Hodkinson

**Affiliations:** 1Botany Department, School of Natural Sciences, Trinity College Dublin, The University of Dublin, College Green, D02 PN40 Dublin, Ireland; fullere@tcd.ie (E.F.); anna.tiley@afbini.gov.uk (A.M.M.T.); hodkinst@tcd.ie (T.R.H.); 2Agri-Food and Biosciences Institute, 18a Newforge Lane, Belfast BT9 5PX, UK; Archie.Murchie@afbini.gov.uk

**Keywords:** ash sawflies, *Fraxinus* spp., Republic of Ireland, *Tomostethus nigritus*

## Abstract

**Simple Summary:**

Ash sawfly, *Tomostethus nigritus*, larvae feed on the leaves of *Fraxinus excelsior*. In the last 20 years, outbreaks of these insects have become more common, and they cause severe defoliation of the tree crown. This pest is native to Europe, and it was recorded for the first time on the island of Ireland in Belfast, Northern Ireland (UK) in 2016. This article is the first report of *T. nigritus* in the Republic of Ireland (IE). Damaged leaves and larvae were observed on ash trees in Co. Kildare in summer 2021. The larvae were collected and then identified using molecular analyses. Similar damaged leaves were observed on trees in Co. Dublin, which showed mild defoliation of the crown. However, severe defoliation of ash trees has also been recorded in Co. Meath in 2021. No control treatments are available against the insects, and little is known about the cause of the outbreaks. Severe outbreaks by *T. nigritus* could further affect the delicate situation that the Irish ash tree population is currently facing caused by another severe antagonist of *Fraxinus*, the ash dieback pathogen *Hymenoscyphus fraxineus*.

**Abstract:**

This is the first report of the ash sawfly, *Tomostethus nigritus*, in the Republic of Ireland. We observed defoliated leaves of *Fraxinus excelsior* L. and *T. nigritus* larvae at a forestry plantation in Co. Kildare. Morphological observation of the larvae and DNA analysis using mitochondrial COI barcoding confirmed the identification of this pest of ash.

## 1. Introduction

*Tomostethus nigritus* (Fabricius, 1804) is a Euro-Siberian species that is distributed throughout many parts of Europe [1]. This species was seldomly associated with severe defoliation of ash trees (*Fraxinus excelsior*) and it was generally present in low densities in urban areas in Western Europe [1,2]. However, severe outbreaks have been intensifying over the last 30 years and severe defoliation of ash trees has been observed in countries such as Austria (1974, 1977, 1999) [3,4]; Belgium (2016 onward) [2]; Croatia (1997–2001) [5]; Czech Republic (1965, 1999–2000) [6,7]; Finland (2015 onwards) [8]; Germany (1993, 1994) [9]; Italy (North Italy: Friuli Venezia Giulia (2000–2009), Lombardia (1980–2007), Veneto (2012-?), Trentino Alto Adige (2008–2009)) [10,11]; Norway (1990–1995) [12]; Republic of Moldova (2018, 2019) [13]; UK (London (1993) [14], Manchester (2010) [15], Telford (2014) [15], Scotland (2014) [15], Northern Ireland (2016 onwards) [1]); and Ukraine (2002, 2012–2014) [16]. Adults are visible around mid-April/May and they deposit their eggs within the ash leaves. Once the larvae hatch, they eat the leaves until middle summer and then they migrate to the soil where they pupate and survive winter [1] before emerging as adults the following spring. There is a single generation per year.

This pest was previously recorded to only attack *F. excelsior* species, however, a new report by Verheyde et al. (2019) demonstrated that the ash sawfly can also cause damage on *F. angustifolia* Vahl [2]. *T. nigritus* was first observed on the island of Ireland in Northern Ireland in 2016, where it is still active in the Belfast area [1]. However, until now, *Tomostethus nigritus* has not yet been reported in the Republic of Ireland.

## 2. Materials and Methods 

### 2.1. Biological Material

Damaged leaves and larvae were first observed at the beginning of July 2021 in an ash tree forestry plantation in Carbury, Co. Kildare, Ireland. The ash plantation, which covers an area of 7000 m^2^, is surrounded mainly by a golf club and a spruce plantation. Leaves and larvae samples were collected and stored at 4 °C until subsequent genomic DNA extraction and sequencing in the laboratory. Only larvae and not adults were apparent in June.

### 2.2. DNA Isolation and Analysis

Extraction of the genomic DNA from the larvae was carried out using the CTAB protocol as described by Hodkinson et al. (2007) [17,18]. PCR amplification was conducted using primers for the cytochrome oxidase subunit 1 (COI) gene, a mitochondrial barcode region used for the identification of insects. LCO1490 (5′-GGTCAACAAATCATAAAGATATTGG-3′) and HC02198 (5′-TAAACTTCAGGGTGACCAAAAAATCA-3′) [19] primers were used for amplification. The cycling parameters were modified from Ashfaq et al. (2019) [20] and were as follows: 1.15 min at 94 °C; 35 cycles of 45 s at 94 °C, 1 min at 52 °C, 1 min at 72 °C; and a final cycle of 7 min at 72 °C.

The PCR product was purified using the ExoSAP-IT™ PCR Product Cleanup Reagent (Thermo Fisher Scientific, Waltham, MA, USA). The final products were sequenced in both directions using either the forward and reverse primer used in the PCR and the Sanger Sequencing Service offered by Source Biosciences (https://www.sourcebioscience.com/home (accessed on 13 July 2021)). The forward and reverse trace files were used to assemble a consensus sequence of 658 bp for each sample using Geneious Prime V. 2021.1.1. The consensus sequence obtained was then analysed using nucleotide-nucleotide BLAST (BLASTn) against the National Centre for Biotechnology Information (NCBI) GenBank database (https://www.ncbi.nlm.nih.gov/ (accessed on 9 September 2021)).

### 2.3. Morphological Observation and Imaging

Morphological observations and imaging of the *T. nigritus* samples were obtained using a dissection microscope. In situ photographs of the *T. nigritus* samples and defoliated ash trees were taken with a CANON EOS 100D or 1100D camera and EFS 18–55 mm lens.

## 3. Results

The larvae were observed on regenerated ash tree leaves (sample code: FeL21Du1-3) and on leaves of a 30-years-old ash tree (sample code: FeL21Du4) (Figure 1a). The larvae were present on the underside of the leaf on a branch at a distance of one to two meters from the ground floor (Figure 1b). Only the main vein of the leaf remained intact and attached to the branches due to the feeding of the larvae (Figure 1e,f). The sawfly larvae were olive green in colour and caterpillar-like and matched the description of Mrkva (1965) for *T. nigritus* [6]. There were mixed instars on the leaves but those observed measured approximately 7 mm in length (Figure 1c).

The mitochondrial cytochrome oxidase gene was successfully amplified and sequenced. The resulting sequences were deposited in GenBank (Table 1, GenBank Accession). The amplicon had a length of 658 bp, which matches the predicted amplicon size [19]. The results from the BLASTn analysis demonstrated that the assembled sequences obtained in this study have a similarity of over 99.8% with the highest NCBI match, which was *T. nigritus* (Table 1, in bold). Other queries obtained from the BLASTn analysis have a pairwise similarity of below 92.10% (Table 1, not in bold). These species have a high pairwise similarity with the sequences obtained from this study because they also belong to the sawfly family Tenthredinidae, suborder Symphyta and order Hymenoptera. Combined, these results therefore confirm that the larvae isolated are identifiable as *T. nigritus*.

## 4. Discussion

Co. Kildare is the first location where *T. nigritus* has been confirmed in the Republic of Ireland. The first recorded observation of this pest on the island of Ireland was in Belfast (Northern Ireland) in 2016 [1]. At first, *T. nigritus* caused severe defoliation on a moderate number of trees and since then an increasing number of ash trees have been completely defoliated by the larvae during the subsequent years [1]. Our observations in Co. Kildare were limited to a small number of trees affected by this insect. Similar mild symptoms affecting the top or the side crown of ash trees (Figure 1e,f and Figure A1) have also been observed by ourselves in urban locations (parks and along canals) in Co. Dublin (Figure 2) during 2021. As explained by Meshkova et al. (2017), different microclimates within the crown could influence larvae and shoot development. This could therefore explain why the insects tended to prefer lower or upper leaves of the crown in different geographic locations [21].

Severe defoliation of ash trees was recorded at a woodland in Co. Meath by a local news article in July 2021 (https://www.meathchronicle.ie/2021/07/15/ash-sawflies-eating-way-through-meath/, last accessed on 15 October 2021)) but identity was not confirmed. However, this research is the first to confirm the presence of *T. nigritus* in the Republic of Ireland using a combination of observational and molecular evidence. According to the newspaper source, the insects were present in such high densities that the larvae were falling from the trees “like rain”. The authors speculate that the ash sawflies may have been introduced in Co. Meath in previous years, perhaps transported from Northern Ireland to the Republic of Ireland.

Adult sawflies have been seen clinging to clothing of cyclists and walkers on the Lagan towpath in Belfast and it is plausible that they could be carried on sheltered parts of cars and lorries in the same manner. The spread in the Belfast area certainly follows parts of the road network. The late spring and subsequent late flushing of ash trees, followed by unusually high temperatures in June and July 2021, could have triggered the severe defoliation observed in Co. Meath. As Verheyde and Sioen (2019) suggest, synchrony between the development of the ash sawfly larvae and the host is the main factor which leads to severe defoliation [2]. Future monitoring of the progress of ash sawfly on the Island of Ireland could help to clarify these speculations.

Ash sawfly, *T. nigritus*, were considered a sporadic pest until now, and little is still known about the factors governing outbreaks. Flooding and insect parasitism are believed to act as natural regulation factors of the ash sawfly population [2,5], but parasitoids attacking *T. nigritus* have not yet been found in Ireland. Global warming and subsequent closer synchrony of insect and plant development are considered to be the main cause of recent widespread outbreaks [2,5,16] but the hypothesis remains to be tested.

Severe outbreaks of ash sawflies do not kill the ash tree, which tends to recover. However, consecutive defoliations might result in reduced vigour [21,22]. A weakened tree can also become vulnerable to more severe threats, such as ash dieback disease caused by the fungus *Hymenoscyphus fraxineus* [22]. Ash dieback is already present on the island of Ireland and it is a major threat to the ash tree population [23]. The combination of ash dieback with the outbreaks of ash sawflies could acerbate even further the already precarious situation of the ash tree population in Ireland. It is therefore crucial to continue monitoring plant health on the Island of Ireland. In addition, it is important to remain vigilant of other threats that could be introduced, such as the emerald ash borer (*Agrilus planipennis* Fairmaire), which could annihilate any effort for recovering and maintaining the Irish ash population on the island.

## Figures and Tables

**Figure 1 insects-13-00006-f001:**
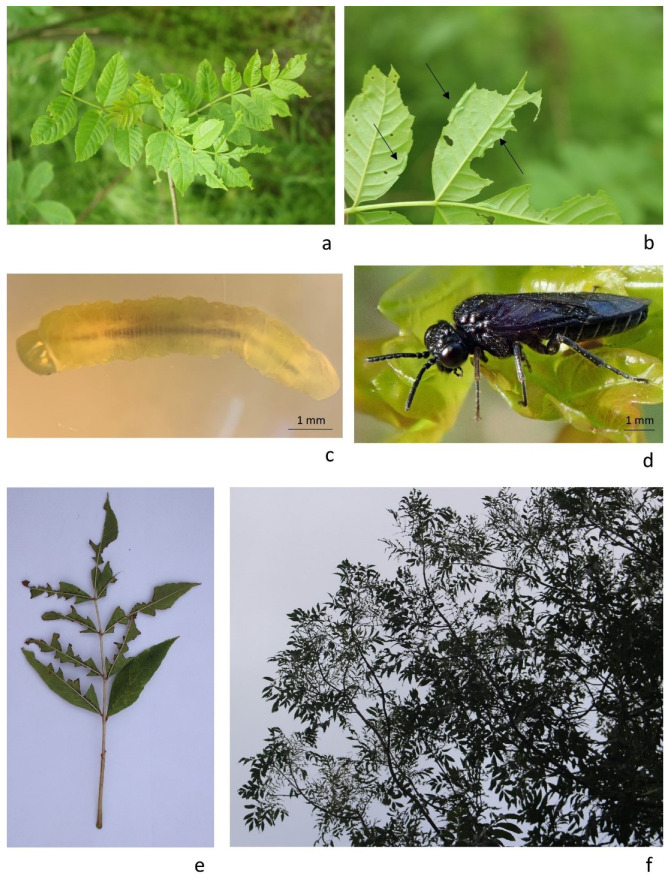
*Tomostethus nigritus* in the Republic of Ireland. (**a**,**b**) Leaves damaged by *T. nigritus* on *F. excelsior* in Co. Kildare and observation of larvae on the underside of the leaves (indicated by the black arrows) (Carbury, Co Kildare, Ireland, 08.VI.21, photo A.M.M. Tiley); (**c**) image of *T. nigritus* larvae (Trinity College Dublin, Dublin, Ireland, 09.VI.21, photo E. Fuller; (**d**) adult of *T. nigritus* observed in Belfast (NI) (Belfast, UK, photo A.K. Murchie); (**e**,**f**) detail of defoliated ash leaves by *T. nigritus* in Co. Dublin. Only the main vein of the leaf remained intact and attached to the branches (Kimmage, Dublin, Ireland, 17.X.21, photo A.M.M. Tiley).

**Figure 2 insects-13-00006-f002:**
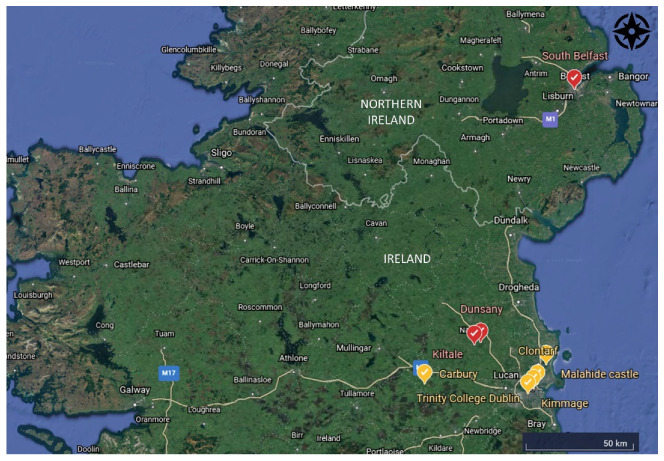
Distribution of observed *Fraxinus* sp. attacked by *T. nigritus* in the Republic of Ireland and Northern Ireland. Trees that shown mild defoliation of the crown are marked in yellow, whereas those with severe defoliation are marked in red.

**Table 1 insects-13-00006-t001:** Summary of nucleotide-nucleotide BLAST (BLASTn) results of samples obtained in this study (FeL21Du1–FeL21Du4) against the NCBI database. BLASTn matches in bold indicate the closest match between the GenBank Accession sequence and Reference Query sequence. BLASTn matches not highlighted in bold have less similarity between the GenBank Accession sequence and Reference Query sequence.

Samples Code	GenBank Accession	Highest Blastn Match	Reference Query	% Pairwise Similarity	% Query Cover
**FeL21Du1**	**OK493754**	** *Tomostethus nigritus* **	**KC972953.1**	**99.85**	**100**
		*Stethomostus fuliginosus*	KC973477.1	92.10	100
		*Ametastegia* sp.	HQ929270.1	90.72	99
		*Monophadnus pallescens*	KC973729.1	90.58	100
		*Eurhadinoceraea amauros*	KC974966.1	90.26	99
		*Empria quadrimaculata*	JN029874.1	90.27	100
		*Paracharactus hyalinus*	KC974771.1	90.26	99
		*Empria rubicola*	JN029873.1	90.12	100
		*Tenthredopsis lactiflua*	KC974941.1	90.09	99
		*Harpiphorus lepidus*	KT964164.1	89.72	99
**FeL21Du2**	**OK493755**	** *Tomostethus nigritus* **	**KC972953.1**	**99.85**	**100**
**FeL21Du3**	**OK493756**	** *Tomostethus nigritus* **	**KC972953.1**	**99.85**	**100**
**FeL21Du4**	**OK493757**	** *Tomostethus nigritus* **	**KC972953.1**	**99.85**	**100**

## Data Availability

DNA sequences are deposited in GenBank under accession numbers OK493754-OK493757 and in the BOLD system under sequences ID GBMNE1845-21-GBMNE1842-21.

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
