# Peer review of "First Report of the Ash Sawfly, Tomostethus nigritus, Established on Fraxinus excelsior in the Republic of Ireland"

_insects, 2021, doi:10.3390/insects13010006_

Round 1
Reviewer 1 Report
The authors describe the first record of the ash sawfly in the Republic of Ireland. Some more information is provided, mainly on its occurrence in Northern Ireland, its status as a pest, and its expected spread on the island. Below are my major and minor comments.
Major comments
1) Microscopic analyses. This is a misleading definition of what has been done, namely simple morphological observations by using not a (light) microscope, but a binocular microscope. Moreover, the observations (two sentences, L80-83) are very simple, vague, and partly false. The term "approximately", two times, should be deleted, since "5 mm" is anyhow too small for the last larval instar -which instar did you used?-, and the number of prolegs is known: the species has eight pairs of prolegs on abdominal segments 2 to 8 and 10. It sounds also strange to emphasize the fact that your "examination demonstrated that the larvae had three pairs of true legs", which simply confirms that they are insects. Thus, overall, adapt all related text parts (from Simple Summary to Results). From my point of view, it seems even unnecessary to mention this method of "microscopic analyses", because the resulting information could have been obtained by sight, or a pocket magnifier.
2) Identification. Change sentence (L16-17) by "The collected larvae were identified using molecular analyses". Why didn't you used, first, identification keys for larvae (Lorenz & Kraus 1957) and/or adults (Benson 1951-1958, Handbooks)? Change subheading (L48), e.g., by "Biological material". Under this part 2.1, you should indicate what has been done with the collected larvae: why were they stored at 4°C? Were they reared? If so, in which conditions? Were adults obtained? Are specimens stored/kept in collections as vouchers? If so, in which institution?
3) Figures 1 and 3. The quality of several pictures is disappointing: Fig. 1 e, f, h and Fig. 3 f have a strange tinge. What are the dark dots/particles in Fig. f, g, h? I guess they may be feces, which should have been removed before taking the pictures. Fig. 1 g seems to have an unfortunate light patch (below, right corner). Legend (L85): "Dissection" is misleading. Overall, both figures contain unnecessary pictures, because being redundant and/or uninformative. Consider rearranging and condensing them into one picture, for instance, by keeping Fig. 1 a, c, i, and Fig. 3 d, e, f.
4) Gene sequencing. Why do you say "approximately" 663 bp (L89)? Table 1 is quite confusing. A legend is missing to explain: the table headings, especially the first ("Sheme .") and two last ones; the meaning of words set in bold; etc. I guess the first GenBank accession "OK493754" is misplaced vertically. It is unclear which sequences are from your study. Note that normally gene sequences are only registered (e.g., in GenBank) once a manuscript is accepted, being mentioned as "XXXXX" until that point. This avoids the situation of orphan accession numbers, that is, not linked to a published work when a journal rejects a manuscript.
Minor comments
Title. Consider deleting "on ash trees".
L17. Rephrase "Similar symptoms..." and/or rearrange the sentences around.
L20. The use of "another" is suboptimal because the sawfly is not a pathogen. You could use "another severe antagonist".
L21. Why "already delicate"? It sounds as if a third (not mentioned) factor would affect ash populations. Or, do you mean already delicate due to the pathogen?
L23-24. Delete "associated with Fraxinus excelsior L." because the "first record" is not about the host plant.
L31. "seldomly"?
L91-92, L96. Set Latin names in italic.
Table 1. Set "sp." not in italic.
L99. Change "Tomostethus" by "T.". By the way, check throughout the text, also for other Latin names, whether you follow the rule to give the full species name (including genus and authorship) only at first mention.
Figure 2. It's difficult to see the geographic link between parts "a" and "b". Consider condensing both into one.
L136, L142. It may be better to use "ash sawfly" in a singular form.
L172. Austara first name is missing.
Author Response
POINT 1) Microscopic analyses. This is a misleading definition of what has been done, namely simple morphological observations by using not a (light) microscope, but a binocular microscope. Moreover, the observations (two sentences, L80-83) are very simple, vague, and partly false. The term "approximately", two times, should be deleted, since "5 mm" is anyhow too small for the last larval instar -which instar did you used?-, and the number of prolegs is known: the species has eight pairs of prolegs on abdominal segments 2 to 8 and 10. It sounds also strange to emphasize the fact that your "examination demonstrated that the larvae had three pairs of true legs", which simply confirms that they are insects. Thus, overall, adapt all related text parts (from Simple Summary to Results). From my point of view, it seems even unnecessary to mention this method of "microscopic analyses", because the resulting information could have been obtained by sight, or a pocket magnifier.
ANSWER 1) The aim of the morphological analysis was to provide general visual observations of the larvae collected. Following the Reviewer’s comments, microscopic analysis has been replaced by morphological observation of the larvae. The text has also been adapted and updated as per the Reviewer’s comments.
POINT 2) Identification. Change sentence (L16-17) by "The collected larvae were identified using molecular analyses". Why didn't you used, first, identification keys for larvae (Lorenz & Kraus 1957) and/or adults (Benson 1951-1958, Handbooks)? Change subheading (L48), e.g., by "Biological material". Under this part 2.1, you should indicate what has been done with the collected larvae: why were they stored at 4°C? Were they reared? If so, in which conditions? Were adults obtained? Are specimens stored/kept in collections as vouchers? If so, in which institution?
ANSWER 2) The larvae were only used for molecular identification. Considering the limited sample size of the larvae, we only had enough samples to run the molecular analysis. No samples were left for further analysis, reared or kept as specimens.
POINT 3) Figures 1 and 3. The quality of several pictures is disappointing: Fig. 1 e, f, h and Fig. 3 f have a strange tinge. What are the dark dots/particles in Fig. f, g, h? I guess they may be feces, which should have been removed before taking the pictures. Fig. 1 g seems to have an unfortunate light patch (below, right corner). Legend (L85): "Dissection" is misleading. Overall, both figures contain unnecessary pictures, because being redundant and/or uninformative. Consider rearranging and condensing them into one picture, for instance, by keeping Fig. 1 a, c, i, and Fig. 3 d, e, f.
ANSWER 3) Excessive or unclear pictures have been removed and Figure 1 and Figure 3 have been condensed into one figure. The word “Dissection” has been removed.
POINT 4) Gene sequencing. Why do you say "approximately" 663 bp (L89)? Table 1 is quite confusing. A legend is missing to explain: the table headings, especially the first ("Sheme .") and two last ones; the meaning of words set in bold; etc. I guess the first GenBank accession "OK493754" is misplaced vertically. It is unclear which sequences are from your study. Note that normally gene sequences are only registered (e.g., in GenBank) once a manuscript is accepted, being mentioned as "XXXXX" until that point. This avoids the situation of orphan accession numbers, that is, not linked to a published work when a journal rejects a manuscript.
ANSWER 4) The DNA sequence length has been corrected to 658 bp. The legend of Table 1 has been updated as follows: “Summary of nucleotide-nucleotide BLAST (BLASTn) results of samples obtained in this study (FeL21Du1 – FeL21Du4) against the NCBI database. BLASTn matches in bold indicate the closest match between the GenBank Accession sequence and Reference Query sequence. BLASTn matches not highlighted in bold have less similarity between the GenBank Accession sequence and Reference Query sequence.”
“Scheme” has been replaced with “Sample code”. The last two headings are usually obtained when a sequence is blasted against a database and they represent the degree of similarity between the query and the reference sequence.
The sequences have been already published in GenBank, without reference to this paper. The authors will therefore keep this comment in mind in the future. We will update the GenBank file with the new information.
Reviewer 2 Report
This is a nice report of a spread of a sawfly Tomostethus nigritus to Republic of Ireland. This expansion of range is expected as the species has been reported from Northern Ireland in 2016. Also, it is well known that this species has shown outbreaks in many areas in Europe, including range expansion. The authors refer to the outbreaks in many countries but fail to mention Finland (evidently, this is my home country) where a massive occurrence of the species started in 2015 and has continued since, starting from the capital area (Helsinki) and then spreading towards the North and West for about 60-70 km. I could find little literature of the spread of this species in Finland, but this might be useful (unfortunately in Finnish): https://jukuri.luke.fi/bitstream/handle/10024/542725/luke-luobio_44_2018.pdf?sequence=1&isAllowed=y
I find this manuscript clear and nice to read. My only comments are minor ones. There is little doubt that the observations really concern T. nigritus as this would have been evident by the larval photographs and the food plant alone. That the authors used NCBI and not BOLD systems to identify their sequences is a bit unfortunate as GenBank is in general less reliable for identifications and lacks lots of more recent COI sequences that have not yet been submitted to GenBank. For this reason, they also fail to notice that there are other species of the same genus (such as T. claripennis) that are clearly closer to T. nigritus than is S. fuliginosus that shows almost 8% divergence to T. nigritus. Actually, the taxonomy of Tomostethus is not clear at all and there are likely undescribed species. Observations on F. angustifolia may actually refer to a different, closely related species (the recent sawfly books by Macek et al and Lacour would shed some more light on this). Thus, this section in the manuscript could be stronger. However, overall, this not a big deal as the species in question evidently is T. nigritus.
In the first paragraph on page 4, the name of T. nigritus is twice without italicization. In table 1, the title of “Gene Bank accession” should be “GenBank accession”. In the same table, the “sp.” after the name Ametastegia should not be italicized.
In the discussion, the authors refer to three papers that apparently state global warming being the main cause behind the recent outbreaks. While this is now a popular explanation for all problems and phenomena, I really wonder what is the evidence supporting this idea. The authors correctly state “this hypothesis remains to be tested”. Indeed, the species occurs to the northern edge of the distribution of ash and a little beyond because ash trees are commonly planted for example in Finland outside the native range of ash. The larvae are able to finish their development long time before the winter and it is hard to believe that e.g. the summer length could limit the distribution. I naturally cannot provide evidence for any alternative explanation, but in sawflies outbreaks are often just normal population dynamics that occur without human intervention. Such outbreak will cease sooner or later regardless of climate warming, usually with help by natural enemies. In Finland, the outbreak seems to be already ceasing in the capital area, but the species seems to be slowly spreading elsewhere. I bet that in ten years we do not see the species anymore anywhere as it has become rare again. The hypothesis by the authors suggesting that we humans transport larvae to new areas makes full sense and sounds likely. I find this a much more likely explanation behind the range expansion that climate warming.
Author Response
POINT 1) This is a nice report of a spread of a sawfly Tomostethus nigritus to Republic of Ireland. This expansion of range is expected as the species has been reported from Northern Ireland in 2016. Also, it is well known that this species has shown outbreaks in many areas in Europe, including range expansion. The authors refer to the outbreaks in many countries but fail to mention Finland (evidently, this is my home country) where a massive occurrence of the species started in 2015 and has continued since, starting from the capital area (Helsinki) and then spreading towards the North and West for about 60-70 km. I could find little literature of the spread of this species in Finland, but this might be useful (unfortunately in Finnish): https://jukuri.luke.fi/bitstream/handle/10024/542725/luke-luobio_44_2018.pdf?sequence=1&isAllowed=y.
ANSWER 1) Thank you for bringing this paper to our attention. The paper suggested has been included at the beginning of the manuscript along with additional locations identified in UK, Italy, Germany and Austria.
POINT 2) I find this manuscript clear and nice to read. My only comments are minor ones. There is little doubt that the observations really concern T. nigritus as this would have been evident by the larval photographs and the food plant alone. That the authors used NCBI and not BOLD systems to identify their sequences is a bit unfortunate as GenBank is in general less reliable for identifications and lacks lots of more recent COI sequences that have not yet been submitted to GenBank. For this reason, they also fail to notice that there are other species of the same genus (such as T. claripennis) that are clearly closer to T. nigritus than is S. fuliginosus that shows almost 8% divergence to T. nigritus. Actually, the taxonomy of Tomostethus is not clear at all and there are likely undescribed species. Observations on F. angustifolia may actually refer to a different, closely related species (the recent sawfly books by Macek et al and Lacour would shed some more light on this). Thus, this section in the manuscript could be stronger. However, overall, this not a big deal as the species in question evidently is T. nigritus.
ANSWER 2) Once again, thank you to bring to our attention the BOLD system. We will make sure that the sequences will be published as well in this database.
POINT 3) In the discussion, the authors refer to three papers that apparently state global warming being the main cause behind the recent outbreaks. While this is now a popular explanation for all problems and phenomena, I really wonder what is the evidence supporting this idea. The authors correctly state “this hypothesis remains to be tested”. Indeed, the species occurs to the northern edge of the distribution of ash and a little beyond because ash trees are commonly planted for example in Finland outside the native range of ash. The larvae are able to finish their development long time before the winter and it is hard to believe that e.g. the summer length could limit the distribution. I naturally cannot provide evidence for any alternative explanation, but in sawflies outbreaks are often just normal population dynamics that occur without human intervention. Such outbreak will cease sooner or later regardless of climate warming, usually with help by natural enemies. In Finland, the outbreak seems to be already ceasing in the capital area, but the species seems to be slowly spreading elsewhere. I bet that in ten years we do not see the species anymore anywhere as it has become rare again. The hypothesis by the authors suggesting that we humans transport larvae to new areas makes full sense and sounds likely. I find this a much more likely explanation behind the range expansion that climate warming.
ANSWER 3) Despite my extensive research into the literature, I was only able to find little information on the causes that are driving the increase of T. nigritus outbreaks in the past 20 years. Verheyde and Sioen (2019) believed that “synchrony in the phenology of the insect with that of its potential host plant” might be the main factor [Verheyde, F.; Sioen, G. Outbreaks of Tomostethus nigritus (Fabricius, 1804) (Hymenoptera, Tenthredinidae) on Fraxinus angustifolia ‘Raywood’ in Belgium. J. Hymenopt. Res. 2019, 72, 67–81, doi:10.3897/jhr.72.38284.]. They also declared that an earlier spring with higher temperatures, possibly due to climate change, might cause the adults to mature earlier and then attack the tree causing severe defoliation due to the synchronization of their phenology. They also observed that trees in an urban environment, were generally targets of these outbreaks all over Europe, while forest outbreaks are still limited to East European countries. However, in our situation, it is still early to be able to declare or observe if there is any difference between outbreaks in the countryside and in the city. I also hope that these phenomena are just temporal as it has been seen also with other species. For example, Stereonychus fraxini De Geer causes recurrently cyclic defoliation of ash trees for approximately 4 to 5 years ceasing thereafter [Drekić, M.; Poljaković Pajnik, L.; Vasić, V.; Pap, P.; Pilipović, A. Contribution to the study of biology of ash weevil (Stereonychus fraxini De Geer). Sumar. List 2014, 138, 387–395].
Reviewer 3 Report
A few comments in the attached pdf.

Author Response
The correct length of the sequence (658 bp) has been updated in the manuscript.
Round 2
Reviewer 1 Report
The manuscript has been adapted adequately.
Minor typos: set Latin name in italic under reference 7.